# GenVarFormer: Predicting gene expression from long-range mutations in cancer

## Abstract

Distinguishing the rare "driver" mutations that fuel cancer progression from the vast background of "passenger" mutations in the non-coding genome is a fundamental challenge in cancer biology. A primary mechanism that non-coding driver mutations contribute to cancer is by affecting gene expression, potentially from millions of nucleotides away. However, existing predictors of gene expression from mutations are unable to simultaneously handle interactions spanning millions of base pairs, the extreme sparsity of somatic mutations, and generalize to unseen genes. To overcome these limitations, we introduce GenVarFormer (GVF), a novel transformer-based architecture designed to learn mutation representations and their impact on gene expression. GVF efficiently predicts the effect of mutations up to 8 million base pairs away from a gene by only considering mutations and their local DNA context, while omitting the vast intermediate sequence. Using data from 864 breast cancer samples from The Cancer Genome Atlas, we demonstrate that GVF predicts gene expression with 26-fold higher correlation across samples than current models. In addition, GVF is the first model of its kind to generalize to unseen genes and samples simultaneously. Finally, we find that GVF patient embeddings are more informative than ground-truth gene expression for predicting overall patient survival in the most prevalent breast cancer subtype, luminal A. GVF embeddings and gene expression yielded concordance indices of $0.706^{\pm 0.136}$ and $0.573^{\pm 0.234}$, respectively. Our work establishes a new state-of-the-art for modeling the functional impact of non-coding mutations in cancer and provides a powerful new tool for identifying potential driver events and prognostic biomarkers.

## 1 Introduction

The completion of the Human Genome Project marked a turning point in biology, revealing that less than 2% of the human genome encodes proteins, leaving the remaining 98%—the non-coding genome—a mystery (Piovesan et al.). Over time, it has become clear that much of this non-coding DNA plays critical roles in regulating gene expression, influencing key biological processes that shape cellular identity and state. Non-coding mutations can alter these regulatory mechanisms by disrupting transcription factor (TF) binding sites (Carrasco Pro et al.), modifying chromatin accessibility (Sundaram et al.), or affecting genome organization (Katainen et al.).

In the context of cancer, which is driven by selective pressures distinct from heritable human evolution, non-coding mutations are particularly intriguing. The vast majority of these mutations are likely to be "passengers," with little impact on tumor fitness. However, a small subset of "driver" mutations are thought to confer a selective advantage and drive carcinogenesis. The challenge lies in distinguishing these rare drivers from the background noise of passengers (Carter et al.).

The increasing availability of whole-genome sequencing data from thousands of cancer samples (Priestley et al.; Aaltonen et al.; Sosinsky et al.) has created new opportunities to investigate the landscape of non-coding mutations in cancer. Non-coding driver mutations must have a biological effect in order to improve tumor fitness. As the primary function of non-coding DNA is gene regulation, it is likely that non-coding driver mutations affect gene expression. As a result, *modeling the effects of non-coding mutations can provide strong evidence for whether they are a driver and link their function to specific genes, revealing how they contribute to tumor fitness.*

Three challenges stand out when building models of gene expression in cancer. First, mutations that affect gene expression can be millions of base pairs (Mbp) away from the gene they affect (Tjalsma et al.). Second, somatic mutations can be relatively infrequent, with an average of 6 mutations per Mbp in non-pediatric cancers (Poulsgaard et al.). Finally, because the space of possible mutations is so large, almost all observed mutations are unique to each tumor. Prior work has tried to address these challenges by using lasso models on so-called mutation "hotspots" (Zhang et al.; Soltis et al.; Pudjihartono et al.). Using hotspots can often increase the frequency of features by one to two orders of magnitude by aggregating mutations that are close together or are in a locus with more mutations than expected by chance. While this addresses the first issue by omitting all DNA context, it also necessitates fitting a model for each gene of interest. In non-cancerous samples, the issue of low mutation frequency has been addressed by sequence models trained on common and rare germline variants simultaneously (Drusinsky et al.; Rastogi et al.; Spiro et al.). However, these attempts have been limited to considering variants at most 25 kilobases (kbp) away from the start of a gene since they require contiguous DNA sequences as input.

To address these challenges, we introduce GenVarFormer (GVF), a novel method for learning representations of mutations that affect gene expression in cancer (§3). GVF predicts gene expression from mutations with a correlation across samples over 26 times greater than current approaches (§4.2). Furthermore, to the best of our knowledge, GVF is the first model to generalize to predicting cancer gene expression in unseen genes and samples using mutations alone. We then use GVF to compute patient embeddings and find that, in the luminal A subtype of breast cancer, GVF embeddings are more informative for patient prognosis than ground truth gene expression (§4.3).

## 2 RELATED WORKS

**Models of frequently mutated regions**  In bioinformatics, a common way to identify mutations that affect gene expression in cancer is to predict gene expression from mutation hotspots using lasso regression (Zhang et al.; Soltis et al.; Pudjihartono et al.). This approach is attractive since it can efficiently incorporate mutations spanning millions of nucleotides and using hotspots can increase the frequency of the input features. However, it suffers from several major limitations. First, it is sensitive to the hotspot calling algorithm used and discards potentially informative non-hotspot mutations. Second, these models don't learn generalizable features, which prevents transfer learning and task adaptation. Finally, it requires fitting models per gene, preventing application to novel genes that can be critical cancer drivers, for example gene fusions (Dashi & Varjosalo) and recently evolved *de novo* genes (Xiao et al.). GenVarFormer (GVF) eliminates these issues: it has no dependency on hotspot calling algorithms, uses all available somatic mutations, generates informative embeddings for downstream tasks, and is a pan-gene model that generalizes to unseen genes.

**DNA sequence-to-function models**  DNA sequence-to-function models for gene expression, which typically apply convolutional neural networks (CNNs) or transformers to one-hot encoded DNA, have recently scaled to contexts of up to 1 Mbp and achieved high accuracy across genes (Avsec et al.). However, these models are not trained on paired human genetic variation and gene expression, and evaluations consistently find they fail to predict expression differences between individuals (Huang et al.; Sasse et al.). While several groups have trained or fine-tuned models specifically on genetic variation to address this gap (Drusinsky et al.; Rastogi et al.; Spiro et al.), the computational expense has limited them to input contexts of only 49 kbp. This narrow context window severely restricts model performance for two key reasons. First, it is too small to capture sparse functional variation. For example, with an average of only 6 somatic mutations per Mbp in cancer (Poulsgaard et al.), a 49 kbp window is unlikely to contain a relevant signal. In the dataset we used, we find that for 88% of genes, a 49 kbp window contains no mutations, making accurate prediction impossible. Second, no model trained on genetic variation has demonstrated generalization to unseen genes. This is a critical failure, as the biological processes governing gene regulation—such as transcription factor binding to promoters and enhancers—are fundamentally shared across the human genome. GVF uses a DNA context window of 16 Mbp—over 340 times larger than prior work—to model the impact of sparse genetic variation and generalizes to unseen genes.

**Vector representations of somatic mutations**   Several efforts have been made to learn representations of somatic mutations for tasks such as cancer type classification and patient survival prediction (Kim et al.; Gupta et al.; Sanjaya et al.; Anaya et al.). Most focus exclusively on coding mutations to either identify them as driver mutations or predict tumor-level phenotypes such as cancer type. For instance, Mut2Vec employs a word2vec-inspired model to embed genes based on their co-occurrence patterns in patient mutation profiles, evaluating whether the representations are informative for finding driver mutations (Kim et al.). Anaya et al. is the most related to our work by using a similar mutation input structure, but demonstrates a multiple-instance learning framework to predict tumor type and microsatellite status, both of which are tumor-level phenotypes. Like other methods, it is also restricted to coding mutations and does not incorporate variant allele frequency (VAF), which can serve as a proxy for what fraction of cells in a tumor have acquired the mutation (Castro et al.). Overall, these methods are less likely to reveal insights about the biological function of mutations because they are trained to predict tumor-level phenotypes that are too high-level to reflect specific biological processes. To this end, several of these methods also bin the genomic position of mutations to the nearest megabase, increasing the representational similarity of mutations but further obfuscating their function. In contrast, GVF is the first model designed to learn representations of non-coding somatic mutations by training on the fundamental biological task of predicting gene expression, dramatically increasing the number of instances the model can learn from and enabling it to learn functionally relevant representations.

## 3 GENVARFORMER

We formulate predicting gene expression from mutations as a regression task where each instance is a particular gene and sample. The inputs are mutations from a 16 Mbp window centered on the gene transcription start site (TSS) along with gene-identifying information. A schematic of the full architecture is shown in Fig. 1. We define mutations with the following properties:

- ALT: the DNA sequence of the mutation, the alternative allele relative to the reference genome.

- ILEN: the indel length of the mutation. Negative for deletions, zero for substitutions, and positive for insertions. All mutations are left normalized (Tan et al.) and atomized such that all substitutions are single-base substitutions and the first nucleotide of an indel corresponds to the reference genome.

- VAF: variant allele fraction, the proportion of sequencing reads that have the mutation in a sample. This provides information about intratumoral heterogeneity and the prevalence of the mutation in a sample.

- Flanking DNA: 32 bp from both the 5' and 3' ends of the mutation.

- POS: the position of the mutation relative to the length of the chromosome it is on.

We then encode each set of input mutations as a sequence of vectors $\mathbb{M} = \{\boldsymbol{m}_0, ..., \boldsymbol{m}_n\} \in \mathbb{R}^d$. More specifically, for each mutation the ALT is encoded via a single layer transformer followed by mean pooling. The ILEN and VAF are stacked into a 2-dimensional vector and projected into $\mathbb{R}^d$. The flanking DNA sequences are concatenated, pass through a shallow convolutional neural network (CNN), and mean pooled; in our experiments we use a randomly initialized ConvNova (Bo et al.) module for this. This yields three $d$-dimensional vectors that are summed together to get each of the mutation vectors $\mathbb{M}$. Finally, to help prevent overfitting, each mutation's position is made relative to the gene's start position and rounded to the nearest hundreds place. The amount of rounding balances between making mutations less unique and the informativeness of a mutation's position relative to a gene. The positions of the mutations are used in subsequent transformer layers for rotary positional embeddings (Su et al.).

Gene specific features are also included in the input since only 6 mutations occur per Mbp on average (Poulsgaard et al.). If GVF only used mutations and their flanking DNA as input, this could make it impossible to identify the gene being predicted when mutations are mostly distal from the gene. In order to ensure these gene specific features are generalizable to unseen genes, we use features derived from DNA sequence. In practice, this can be the gene promoter and/or coding DNA, and in our experiments we use embeddings from Borzoi (Linder et al.), a sequence-to-function model.

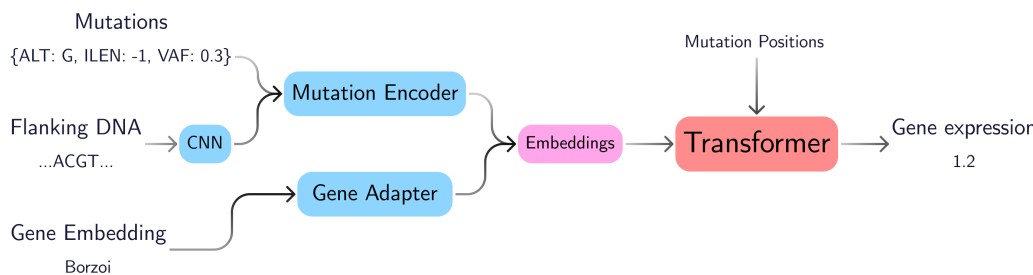

Figure 1: Schematic of GenVarFormer (GVF). Mutations from a 16 Mbp window around a gene are given as input. Each mutation's information including the ALT, ILEN, VAF, and flanking DNA are transformed into an $\mathbb{R}^d$ vector, as well as a gene embedding from Borzoi (Linder et al.) and all are summed together to get embeddings. These pass through a transformer along with mutation positions that are used for rotary positional embeddings (Su et al.). Finally, the embeddings for each mutation are mean pooled to a single vector and projected to predict gene expression.

As the gene adapter, we use a randomly initialized ConvNova (Bo et al.) module followed by mean pooling to compute a $d$-dimensional vector, which is added to all mutation vectors $\mathbb{M}$. In our case where gene embeddings are from a pretrained model, we multiply them by a learnable parameter initialized to a small value, e.g. $10^{-6}$. This prevents the gene embedding from dominating the inputs at the start of training.

After encoding the mutations and gene-specific features into $\mathbb{M}$, they are passed through a transformer, mean-pooled, and projected to a scalar value to predict gene expression. For the loss function, we use gradient aligned regression (Zhu et al.) as minimizing pairwise distances has been shown to be critical in prior work predicting gene expression outside of cancer (Drusinsky et al.; Rastogi et al.; Spiro et al.). We additionally follow prior work and ensure that each batch seen during training exclusively consists of instances from the same gene.

### 3.1 TECHNICAL ADVANCES

Several technical challenges emerged while building GVF. First, we observed that the distribution of mutations per instance in the dataset was approximately Zipf-distributed (Fig. 2A). This meant that conventional padding strategies would lead to batches consisting of almost 100% pad tokens and make GVF infeasible for practical use (Fig. 2B). We thus implemented GVF using PyTorch (Paszke et al.) nested tensors to eliminate the need for padding during training and inference. However, even without padding we found that naive random sampling would cause out-memory-errors, as the number of mutations per batch was not limited by the batch size. This was remedied by implementing a bin packing sampler that ensured no batch contained more than a set number of mutations (Fig. 2C). As an added benefit, this sampling strategy also maximized the number of mutations per batch that could fit into memory and boosted GPU utilization. Finally, unlike natural language, mutations are not regularly spaced and to our knowledge, there is no implementation of rotary positional embeddings (RoPE) (Su et al.) that supports arbitrarily positioned tokens. We thus implemented a new RoPE Triton kernel for arbitrarily positioned tokens, building on the implementation from FlashAttention (Dao). Similarly, we used FlashAttention for all attention operations in GVF as it was the only implementation we found with a robust and performant forward and backward pass for nested tensors.

After overcoming these technical challenges, we were able to apply GVF to the full dataset without any issues. To demonstrate the gains in efficiency of using GVF over conventional biological sequence models, we benchmarked the throughput of GVF against Flashzoi (Hingerl et al.), a FlashAttention-enhanced version of Borzoi (Linder et al.), with matched window sizes of 524,288 bp. For GVF, we set the maximum number of variants per batch to 32,768, and for Flashzoi we used a batch size of 8—the largest power of 2 that would fit into GPU memory. We found that GVF was over 1,170 times faster than Flashzoi while using less GPU memory (Fig. 2D).

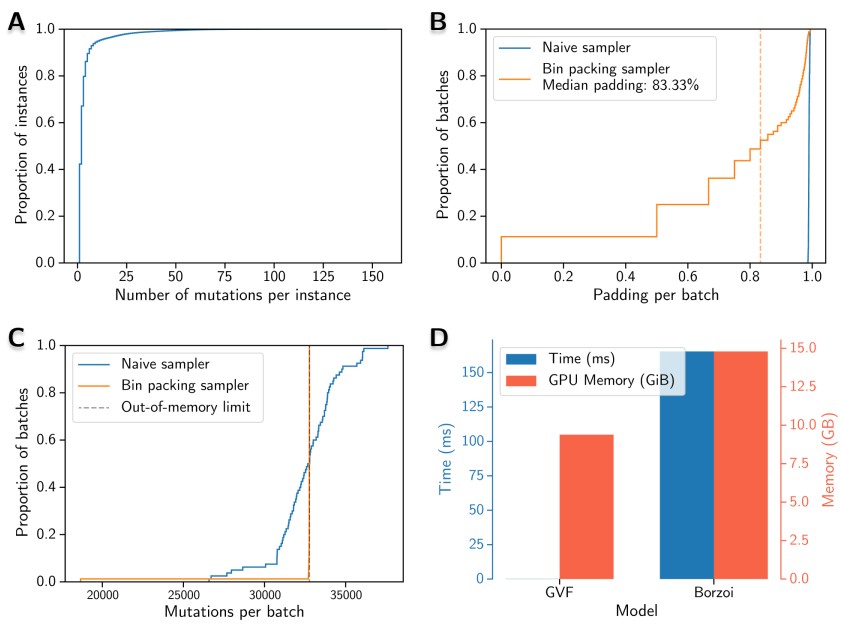

Figure 2: Motivation and benefit of using nested tensors throughout GVF. A) The distribution of mutations per instance in the dataset. This is approximately Zipf-distributed. B) The Zipf-distribution of mutations per instance causes extremely high amounts of padding per batch, event with the bin packing sampler. Dashed orange line is the median padding per batch with the bin packing sampler. C) Naive random sampling with a fixed batch size leads to out-of-memory errors since the number of mutations in a batch varies. The bin packing sampler strictly respects memory limits and increases GPU utilization by maximizing the the number of mutations per batch. D) Comparison of the average inference time per instance of GVF and Borzoi, as well as their peak memory usage. GVF is over 1,170 times faster at computing predictions.

## 4 EXPERIMENTS

We conduct two main experiments with GenVarFormer (GVF). First, we trained and evaluated it for predicting gene expression in tumors from somatic mutations, assessing its generalization to unseen samples, unseen genes, and simultaneously unseen samples and genes. Second, we computed patient embeddings with GVF and used linear probes to assess their clinical utility by predicting patient progression-free and overall survival, PAM50 subtype (Parker et al.), and early vs. late tumor stage.

### 4.1 DATA, SPLITTING, AND TRAINING

We obtained paired whole-genome and bulk RNA sequencing for 864 breast cancer samples from The Cancer Genome Atlas (TCGA) (Weinstein et al.). Since solid tumor biopsies are almost always a mixture of cancer and non-cancer cells, bulk RNA-seq measures a blend of cancer and non-cancer cells. This confounds the task of predicting cancer gene expression from mutations. Using the bulk expression encourages the model to learn how mutations in cancer cells affect gene expression in other cell types, which is mediated by dramatically different biological processes than within-cell gene regulation. To focus on cancer gene regulation, we used InstaPrism (Hu & Chikina) to estimate the cancer gene expression (*in silico* purification), removing the amount attributable to other cell types. We then generated splits of unseen samples (US), unseen genes (UG), and both (USG) for testing, validation, and training. We split the genes by chromosome such that the test and validation splits each had approximately 10% of the genes with any non-zero measurements. We then used 5-fold cross-validation across samples within the training split for hyperparameter tuning. Similar to prior work (Zhang et al.; Soltis et al.; Pudjihartono et al.), we finally regressed out the top 10 principal components of gene expression to remove batch effects and z-scored the residuals, respecting the 5 training folds, validation, and test splits. For training, we subset the dataset to genes

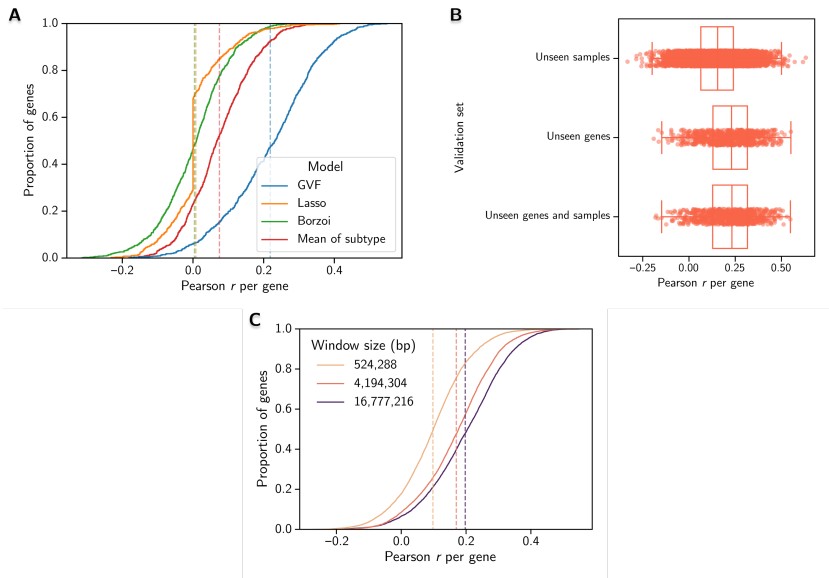

Figure 3: A) Performance of GVF, lasso models using mutation hotspots, Borzoi, and the mean gene expression of each breast cancer subtype. GVF performs over 26 times better than lasso models using the same input modality. Dashed, colored lines are the mean of each model's performance. B) GVF demonstrates generalization in all three validation sets, most notably in unseen genes and samples with an average Pearson $r$ of 0.219. C) As a result of improved scalability, GVF was trained with 16 Mbp windows which yielded double the performance of 524 kbp windows.

with a mean, purified expression greater than 1 (in $\log(\text{TPM} + 1)$). After training to predict gene expression, we used 3-fold nested cross validation for benchmarking on downstream tasks.

## 4.2 Predicting gene expression

To benchmark GVF, we computed the performance of three baselines: lasso models using hotspots called with the algorithm described in Zhang et al., predictions from Borzoi (Linder et al.) for all breast and breast cancer cell lines it was trained on, and the mean expression of the training data for each gene and subtype. Borzoi outputs 32-bp resolution tracks, so we follow (Linder et al.) and summed the predictions across exons to compute gene-level predictions. We did not fine-tune Borzoi due to computational constraints. We quantified performance by computing the Pearson correlation across samples for each gene and then computing the average of each gene's correlation. We found that Borzoi yielded the lowest average correlation at 0.0043, followed by the lasso models at 0.0081, the mean subtype at 0.0749, and GVF at 0.2187. Borzoi's low performance is unsurprising given that it was never trained on genetic variation and this result is consistent with reports evaluating similar models in non-cancerous tissue (Huang et al.; Sasse et al.). The lasso models represent a typical approach taken in literature (Zhang et al.; Soltis et al.; Pudjihartono et al.), which GVF outperforms by over 26-fold (Fig. 3A). Notably, GVF achieves this while generalizing to unseen genes and samples (Fig. 3B), which has yet to be reported for any existing biological sequence model (Drusinsky et al.; Rastogi et al.; Spiro et al.). We include the performance of the average expression per gene and subtype from the training data as a strong but simple baseline to evaluate whether GVF could be learning this straightforward relationship between expression and subtype. Finally, we conducted an ablation study with window sizes of $2^{19}$, $2^{22}$, and $2^{24}$, finding that GVF improved in performance with increasing window size, achieving roughly double the performance with a 16 Mbp vs. 524 kbp window (Fig. 3C).

## 4.3 Evaluating GenVarFormer representations for clinical utility

To evaluate whether GVF could generate clinically informative patient-level representations, we first extracted embeddings from every layer in the model for every instance in the dataset. Each instance

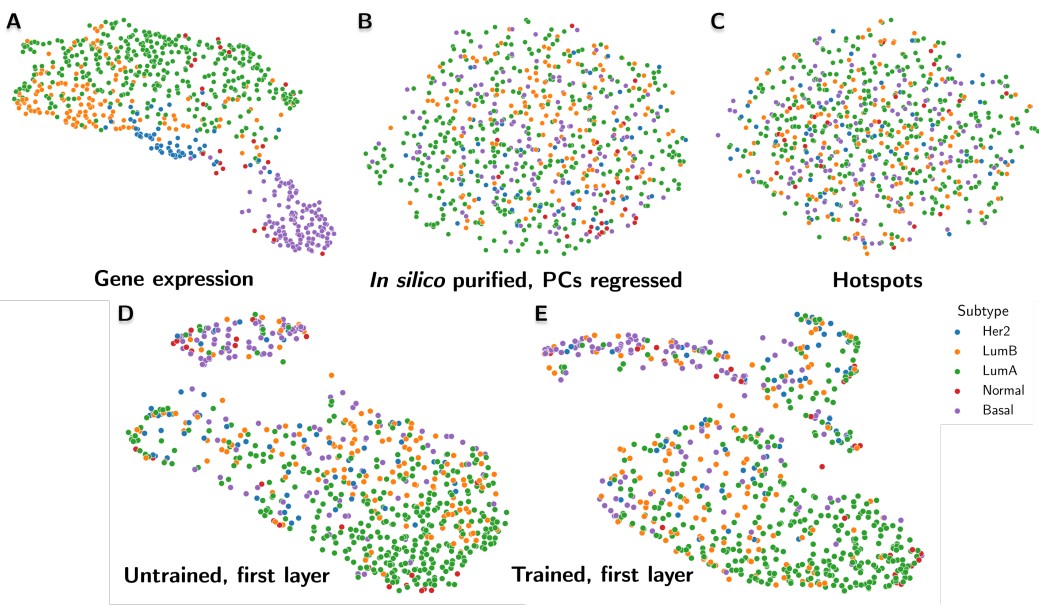

Figure 4: GVF patient representations display structure not seen in either outputs (B) or inputs (approximated by C). Tumors colored by subtype and projected via UMAP of A) gene expression, B) *in silico* purified gene expression with the top 10 PCs regressed, C) mutation hotspots, D) random projections from the mutation encoder/first layer of an untrained GVF model, and E) embeddings from the first layer of the trained GVF model. Random projections (D) are expected to show high-level patterns of mutations as a form of dimensionality reduction.

and layer yielded as many $d$-dimensional vectors as there were mutations, so we mean pooled the mutations to get gene-level embeddings. Finally, we concatenated these gene embeddings to get patient-level embeddings. We then generated UMAPs of gene expression, purified expression with the top 10 PCs regressed (the target that GVF was trained to predict), mutation hotspots, and embeddings from the mutation encoder/first layer of an untrained and trained GVF model (Fig. 4). As a case study, we then colored these projections by PAM50 (Parker et al.) subtype, an important prognostic biomarker for breast cancer. PAM50 subtypes are defined by a nearest-centroid classifier, Prediction Analysis of Microarray (PAM) (Tibshirani et al.), using the expression of 50 genes that were chosen to maximize progression-free risk stratification. Thus, it is expected that gene expression would largely reflect PAM50 subtypes as in Fig. 4A. After *in silico* purification and regressing out the first 10 principal components, subtype-specific patterns in gene expression are largely lost (Fig. 4B). Likewise, mutation hotspots do not appear to strongly correspond to subtype, nor show much overall structure (Fig. 4C).

Since breast cancer subtypes display distinct patterns of mutations (Perry et al.), we hypothesized that random projections of patients from an untrained GVF mutation encoder/first layer would also display some degree of clustering by subtype. The UMAP of patient embeddings from an untrained mutation encoder suggest this is true, as patients roughly stratify by subtype (Fig. 4D). This also shows that even random projections from GVF's mutation encoder encode substantially more information than hotspots. Note that the random projections from the untrained mutation encoder only incorporate information about the DNA sequence resulting from mutations, not the positions, substitutions, or indels that occurred. The UMAP of GVF's trained mutation encoder showed a similar, but finer-grained, substructure of patients (Fig. 4E). We quantified the informativeness of hotspots, random patient projections, and trained patient embeddings in the next experiment.

After this case study of PAM50 subtypes in the latent space, we focused on fitting linear probes to predict clinical annotations: progression-free and overall survival as well as PAM50 subtype and early vs. late stage by binning tumor stage into I-II and III-IV 1. We also fit and evaluated survival within each subtype since the two are strongly linked; subtype is a key biomarker that helps guide

treatment. We used mutation hotspots and an untrained GVF (i.e. no pre-training) as baselines for direct comparison to GVF. As a point of reference, we also evaluated advantaged feature sets: gene expression, *in silico* purified gene expression, and gene copy number. These features are not available to GVF and take advantage of either having RNA-seq available or knowledge of gene coordinates. As a result these feature sets also have 64 times fewer dimensions, corresponding to the dimensionality of GVF gene embeddings. For each combination of feature set and survival task, we projected the features onto their principal components and fit a Cox proportional hazard model to the projections. The number of principal components—and best layer of the model when applicable—was selected using the inner cross validation loop. For the non-survival tasks, we fit logistic regression models with an L2 penalty selected by cross validation. For several tasks, we observed that no mutation-based feature set yielded a score that was more than 1 standard deviation away from random, suggesting that with only 864 samples there may not have been enough data to fit meaningfully performant models. However, trained GVF embeddings were the most performant for every task where we could fit a mutation-based model that was substantially better than random.

Table 1: Performance for predicting overall (OS) and progression-free survival (PFS), PAM50 subtype, and early/late stage cancer. Early/late is defined as Stage I-II and Stage III-IV. Survival was also evaluated in each subtype. Values and standard deviations for survival are concordance indices, and for classification are for the area under the receiver-operating characteristic, which is one-vs-rest macro averaged for PAM50 subtype. Tasks where no mutation-only feature sets yielded performance greater than 1 standard deviation away from random performance were omitted. Advantaged features generally have a much higher signal-to-noise ratio (SNR) than mutations and are included for reference. Bold and underlined entries indicate the best and second-best non-random scores among mutation-only feature sets. Expr: gene expression. Pure: *in silico* purified expression. CN: gene copy number. Hotspots: mutation hotspots. GVF-R: randomly initialized, untrained GVF. PAM50: whether the features were subset to the PAM50 genes or not.

| Features | OS:LumA | OS:Basal | PFS:LumA | PAM50 | Early/Late |
|---|---|---|---|---|---|
| Sample size | $n = 437$ | $n = 155$ | $n = 437$ | $n = 864$ | $n = 864$ |
| *Advantaged Features* | | | | | |
| Expr | 0.53 ±0.28 | 0.77 ±0.18 | 0.63 ±0.1 | 0.98 ±0.0 | 0.61 ±0.04 |
| Expr, PAM50 | 0.57 ±0.23 | 0.54 ±0.33 | 0.51 ±0.11 | 0.99 ±0.0 | 0.56 ±0.04 |
| Pure | 0.57 ±0.21 | 0.54 ±0.3 | 0.55 ±0.15 | 0.99 ±0.0 | 0.63 ±0.02 |
| Pure, PAM50 | 0.42 ±0.28 | 0.64 ±0.3 | 0.55 ±0.13 | 0.98 ±0.0 | 0.54 ±0.07 |
| CN | 0.51 ±0.2 | 0.45 ±0.28 | 0.51 ±0.19 | 0.9 ±0.01 | 0.58 ±0.03 |
| CN, PAM50 | 0.58 ±0.21 | 0.8 ±0.18 | 0.43 ±0.12 | 0.88 ±0.01 | 0.55 ±0.03 |
| *Mutations Only (fair comparison to GVF)* | | | | | |
| Hotspots | 0.47 ±0.24 | 0.31 ±0.26 | 0.46 ±0.09 | 0.61 ±0.03 | 0.52 ±0.03 |
| GVF-R | 0.64 ±0.22 | 0.43 ±0.29 | 0.66 ±0.16 | 0.72 ±0.02 | 0.5 ±0.05 |
| GVF-R, PAM50 | 0.63 ±0.15 | 0.68 ±0.22 | 0.61 ±0.13 | 0.67 ±0.02 | 0.55 ±0.03 |
| GVF | 0.58 ±0.17 | 0.53 ±0.27 | **0.67 ±0.14** | **0.82 ±0.02** | **0.56 ±0.01** |
| GVF, PAM50 | **0.71 ±0.14** | **0.71 ±0.21** | 0.65 ±0.14 | 0.78 ±0.03 | 0.56 ±0.03 |

## 5 CONCLUSION

We have presented GenVarFormer (GVF), a transformer-based model that expands our ability to functionally interpret non-coding mutations in cancer by predicting their effect on gene expression. By selectively modeling only mutations and their local sequence context, GVF efficiently models cancer gene regulation across up to 16 million base pairs. This context window is 340 times longer than the maximum length used by current personalized sequence-to-function models (Drusinsky et al.; Rastogi et al.; Spiro et al.) and runs over 1,170 times faster than a state-of-the-art sequence

model, Flashzoi (Hingerl et al.). GVF also demonstrates a 26-fold increase in correlation across samples compared to traditional hotspot-based methods and is the first model of its kind capable of generalizing across both unseen genes and patients simultaneously. Finally, GVF patient embeddings proved more informative for predicting survival in luminal A breast cancer than the gene expression data used for training.

One of the most important challenges in cancer is to identify the driver mutations that boost cancer cells' ability to grow, evade the immune system, and otherwise improve their evolutionary fitness (Hanahan). As fundamental contributors to tumor fitness, driver mutations are invaluable biomarkers for patient prognosis and treatment and drug target discovery (Ostroverkhova et al.). Non-coding driver mutations must have an effect relevant to tumor fitness, and the primary function of non-coding DNA is gene regulation. Models of gene expression from mutations can therefore help discern which mutations meet this condition to be a driver mutation. Several challenges impede this task, as non-coding mutations can be multiple megabases away from the gene(s) they affect and they occur with very low frequency across sequence length and patients. GVF overcomes these challenges, offering a powerful new tool to quantify the consequences of non-coding mutations on gene expression and, as a result, prioritize non-coding driver mutations.

Several future directions stand out. Germline variants are known to influence cancer gene expression (Li et al.), so incorporating them may improve predictive performance and reveal novel interactions between germline and somatic mutations. Extending to germline variants would also enable applications beyond cancer, where GVF may enable a more precise understanding of rare variants in genetic disease. We also did not delve into coding mutations in this work, and Anaya et al.'s findings suggest that indicating each mutation's position in any overlapping reading frames would be necessary to enable discrimination between classes of coding mutations. Despite these limitations, our model established a new state-of-the-art for predicting gene expression from mutations in cancer. GenVarFormer helps to provide a path toward representing patient genomes more holistically, moving beyond a narrow focus on recurrent hotspots and coding mutation biomarkers.

## 6 ETHICS

Real patient data was used for this study. As such we conducted all work consistent with the data access policies set by the data distributor, Genomic Data Commons.

## 7 REPRODUCIBILITY

To ensure transparency and reproducibility, model code and weights will be publicly released. Datasets used for this work are available at the Genomic Data Commons Portal.

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

# A Appendix

## A.1 Hyperparameters

To train GVF, we conducted a minimal, manual hyperparameter search using the 5 training folds. We ultimately chose to use AdamW (Loshchilov & Hutter, a) with a learning rate of $10^{-4}$, $\beta$s of (0.9, 0.95), and $\epsilon$ of $10^{-8}$. We also used a cosine learning rate scheduler (Loshchilov & Hutter, b) from TIMM (Wightman) starting and ending the learning rate at $10^{-5}$ with 1 warm up epoch and complete decay after 3 epochs, with no cycles. The final GVF model had 4 transformer layers and roughly 1M parameters.

## A.2 LLM Usage

We used LLMs for drafting, polishing language, and literature search.

