# OpenReview forum: "GenVarFormer: Predicting gene expression from long-range mutations in cancer"
_ICLR.cc/2026/Conference — Submitted to ICLR 2026_

### Official Review · Reviewer_Fr92 · 2025-10-27

**Soundness:** 2
**Presentation:** 3
**Contribution:** 2
**Rating:** 4
**Confidence:** 4

**Summary:**

This paper introduces GenVarFormer (GVF), a transformer-based model that predicts tumor gene expression directly from sparse somatic mutations, targeting the long-range, non-coding “driver vs passenger” problem in cancer. GVF considers variants up to 16 Mbp from a gene, hence addressing the limitations of methods only considering proximal variants. Trained on 864 TCGA breast tumors with in silico–purified cancer expression targets, the model is evaluated under splits that test generalization to unseen samples, genes, and both simultaneously. GVF achieves a 26-fold higher average across-sample correlation per gene than hotspot-lasso baselines (0.219 vs 0.008) and also outperforms sequence-to-function models. Architecturally, GVF uses nested tensors, a bin-packing sampler that caps total variants per batch, and a Triton RoPE kernel for arbitrarily positioned tokens, enabling scalable training without padding blow-ups. GVF is reported to be three orders of magnitude faster than Flashzoi at matched context with lower peak memory, and performance roughly doubles when the window expands from ~0.5 Mbp to 16 Mbp. The authors also show that patient embeddings derived from GVF layers capture subtype structure and provide clinically useful signals beyond mutation hotspots.

**Strengths:**

- The authors aim to predict gene expression from sparse somatic mutations across megabase scales, squarely tackling long-range, non-coding drivers with a custom-built model rather than sequence-only surrogates.
- There are some nice tricks in this paper, such nested tensors, a bin-packing sampler, and a Triton RoPE kernel that enable long contexts without padding blow-ups; they also show major speed/memory wins versus Flashzoi at matched context.
- Training on TCGA BRCA samples and testing generalization across unseen samples, unseen genes, and both simultaneously makes the results more convincing than single-split reporting.
- They report significantly higher across-sample correlation per gene than hotspot-lasso, and beat sequence-to-function models.
- Patient embeddings appear to capture subtype structure and even outperform ground-truth expression for overall survival in luminal A, suggesting bona fide translational utility.

**Weaknesses:**

Please refer to questions.

**Questions:**

- I am not sure why the authors are only focusing on somatic mutations. Gene expression is a function of all variants, germline and somatic. In fact, germline mutations are one of the main reasons different people have significantly different prognosis and outcomes. I don’t believe germline mutations can be left for future work in this context.
- All training/eval is on 864 TCGA breast tumors; there’s no external cohort (e.g., CPTAC) or pan-cancer validation, so generalizability is unclear.
- There are many existing tools (AlphaGenome, BigRNA, ChromBPNet, DeepSEA, Enformer, etc) that predict the consequences of genomic variants on chromatin accessibility, histone marks, TF/RBP binding, promoter activity, (alternative) splicing, gene expression, etc. Why are none of them used here? Sure, the cellular context in cancer cells will be different, but the model will be able to learn it.
- Depending on cancer type, a significant number of genes (in particular tumor suppressors) are silenced through hyper-methylation of their promoters (CpG islands). How are absence of epigenomic factors handled in this analysis? On that note, how is the performance breakdown across oncogenes, tumor suppressors, and other gene groups?
- In most cancers, a small number of activated key oncogenes start de-regulating cellular processing with massive consequences, e.g., change in expression of other genes, not because they have somatic mutations, but due to other genes being activated or silenced. What will be GVF's performance in such cases?
- The authors mention that one of the major benefits of their method over others is its generalizability to new genes, such as fusion products. How can this be validated in TCGA data?
- If Borzoi cannot be fine-tuned due to resource constraints, it should not be included. A Pearson correlation of 0.0043, which is most likely not statistically significant, is not very informative.
- A correlation number by itself, especially when it is only 0.22, is not very meaningful. A couple of examples of how GVF can find hallmark genes with significant predicted and observed differential expression will be helpful.
- The authors predict “in silico purified” cancer expression (InstaPrism) and also regress out the top 10 PCs before z-scoring; both steps can remove real signal or introduce target noise/bias. There are genuine differences between subtypes, which seems to be wiped out (Figs. 4A and B).
- The survival analysis results use linear probes and are strongest only in luminal A (where are Luminal B and Her2?); robustness across subtypes and cohorts remains untested. I would also suggest reporting the Cox HRs.

---

> ### Author Response · Authors · 2025-11-26
> **Reply to review (1/5)**
>
> We appreciate the reviewer’s comments that span many interesting directions, including germline-somatic interactions, epigenomic determinants of gene expression, and GVFs utility to study structural variants such as fusion products. We have responded to each weakness/question below.
>
> **I am not sure why the authors are only focusing on somatic mutations. Gene expression is a function of all variants, germline and somatic. In fact, germline mutations are one of the main reasons different people have significantly different prognosis and outcomes. I don’t believe germline mutations can be left for future work in this context.**
>
> We appreciate the reviewer raising this point. We agree that germline variants ultimately need to be included for a comprehensive model of gene regulation in cancer, especially given prior work showing that germline variants affect somatic gene expression in breast cancer and known germline-somatic interactions in various cancers [1,2,3]. However, in the context of finding driver mutations, we reasoned that almost all driver mutations in breast cancer would be somatic rather than germline (germline-somatic interactions being a clear exception). In addition, since evolutionary constraints on germline and somatic mutations differ [4], we reasoned that somatic mutations are likely to have much larger effect sizes than germline variants. Finally, it’s far more computationally tractable to exclusively use somatic mutations. Each individual has roughly 3 orders of magnitude more germline variants than somatic mutations and, as a transformer, GVF’s computational requirements scale quadratically with the number of variants/mutations. As a result, we started with a model that focuses on somatic mutations. We will mention this future direction in the discussion.
>
> [1] Q. Li et al., “Integrative eQTL-Based Analyses Reveal the Biology of Breast Cancer Risk Loci,” Cell, vol. 152, no. 3, pp. 633–641, Jan. 2013, doi: 10.1016/j.cell.2012.12.034.
>
> [2] Y. Liu, A. Gusev, and P. Kraft, “Germline Cancer Gene Expression Quantitative Trait Loci Are Associated with Local and Global Tumor Mutations,” Cancer Res, vol. 83, no. 8, pp. 1191–1202, Apr. 2023, doi: 10.1158/0008-5472.CAN-22-2624.
>
> [3] N. Giannareas et al., “Extensive germline-somatic interplay contributes to prostate cancer progression through HNF1B co-option of TMPRSS2-ERG,” Nat Commun, vol. 13, no. 1, p. 7320, Nov. 2022, doi: 10.1038/s41467-022-34994-z.
>
> [4] I. Martincorena et al., “Universal Patterns of Selection in Cancer and Somatic Tissues,” Cell, vol. 171, no. 5, pp. 1029-1041.e21, Nov. 2017, doi: 10.1016/j.cell.2017.09.042.
>
> **All training/eval is on 864 TCGA breast tumors; there’s no external cohort (e.g., CPTAC) or pan-cancer validation, so generalizability is unclear.**
>
> We thank the reviewer for this suggestion. (Reviewer 2 and 3 had very similar comments so we have written a unified response.) Unfortunately, CPTAC-3 (CPTAC-2 has no WGS) only contains a single breast cancer case with paired WGS and RNA-seq. We will add a discussion on the need for more public, independent datasets with paired WGS and RNA-seq from breast cancer. We will also explore evaluating using data from Hartwig Medical Foundation, the most comparable independent dataset to our knowledge. However, we expect performance to drop since Hartwig consists of almost exclusively metastatic tumors, whereas TCGA (the training data) is almost exclusively primary tumors. Generalization across domains is out of scope for this work.
>
> **There are many existing tools (AlphaGenome, BigRNA, ChromBPNet, DeepSEA, Enformer, etc) that predict the consequences of genomic variants on chromatin accessibility, histone marks, TF/RBP binding, promoter activity, (alternative) splicing, gene expression, etc. Why are none of them used here? Sure, the cellular context in cancer cells will be different, but the model will be able to learn it.**
>
> (continued)

---

> ### Author Response · Authors · 2025-11-26
> **Reply to review (2/5)**
>
> Such tools are sequence-to-function (S2F) models that use contiguous DNA as input. Fine-tuning S2F models for cancer faces several challenges. The maximum sequence length of any published tool is 1 Mbp and there are 1-2 mutations per Mbp in this breast cancer cohort. With ~800 samples, a generous estimate is ~1600 mutations for each gene to learn from. If we assume each affects a different position, ~99.84% of each 1 Mbp input would be constant across the dataset. As the vast majority of positions would be constant, this is likely to have a very low signal-to-noise ratio (SNR) that hampers optimization. Another challenge is constructing accurate sequences for input. Due to potential intratumoral heterogeneity, aneuploidy, and rearrangements, accurate construction of contiguous somatic sequences generally amounts to de novo assembly of tumor genomes. This is generally impossible from WGS alone. Any sequence construction would be a rough approximation and further reduce the SNR. Finally, the amount of compute required to train S2F models scales rapidly with sequence length. This breast cancer dataset consists of ~800 samples and ~10,000 genes. A hypothetical fine-tuning dataset for AlphaGenome would thus consist of ~8 trillion nucleotides, vastly greater the original ~38 billion nucleotides AlphaGenome was trained on and approaching the size of the Evo 2 dataset at 9.3 trillion nucleotides, the largest published to date [1,2].
>
> [1] Ž. Avsec et al., “AlphaGenome: advancing regulatory variant effect prediction with a unified DNA sequence model,” July 11, 2025, bioRxiv. doi: 10.1101/2025.06.25.661532.
>
> [2] G. Brixi et al., “Genome modeling and design across all domains of life with Evo 2,” Feb. 21, 2025, bioRxiv. doi: 10.1101/2025.02.18.638918.
>
> **Depending on cancer type, a significant number of genes (in particular tumor suppressors) are silenced through hyper-methylation of their promoters (CpG islands). How are absence of epigenomic factors handled in this analysis? On that note, how is the performance breakdown across oncogenes, tumor suppressors, and other gene groups?**
>
> We appreciate the reviewer bringing up epigenomic factors. As presented, GVF can only make predictions from mutations, so epigenetic signals that affect gene expression should add noise to the relationship between mutations and gene expression. Since only mutations are used, GVF is modeling the total effect of mutations on gene expression, including any indirect effects that are sufficiently present in the data. Genetic variants have been associated with virtually every epigenetic signal measured, e.g., methylation, chromatin accessibility, transcription factor binding, so it is almost certain that somatic mutations could affect these processes as well. Depending on the degree to which these epigenomic factors (such as tumor suppressor promoter hyper-methylation) are present and associated with mutations, GVF could be picking up on these signals. We would also expect overall performance to improve from incorporating methylation. However the methylation data available in this cohort is relatively inconsistent, with roughly half having only HM27 methylation array data available, and the other half having HM450. Other epigenomic assays (ATAC, ChIP, Hi-C, etc.) are even less common in most public cancer datasets, especially to be paired with WGS and RNA-seq. Nevertheless, it is an important direction to ultimately integrate the genomic, transcriptomic, and epigenomic data that is available and we will add a brief discussion of this. The suggestion to include a performance breakdown across oncogenes and tumor suppressors is also well taken, and we will include this in the final revision. It will be interesting to see where these genes lie in the distribution of performance and whether there is a potential enrichment of functional mutations targeting these genes.
>
> **In most cancers, a small number of activated key oncogenes start de-regulating cellular processing with massive consequences, e.g., change in expression of other genes, not because they have somatic mutations, but due to other genes being activated or silenced. What will be GVF's performance in such cases?**
>
> (continued)

---

> ### Author Response · Authors · 2025-11-26
> **Reply to review (3/5)**
>
> We agree with the reviewer that oncogene activation often leads to transcriptome-wide changes in gene expression. This is particularly well known in breast cancer with the PAM50 gene expression subtypes being closely associated with functional coding mutations in ERBB2, ESR1, PGR, or a lack of mutations in all three [1]. We regress out the top 10 gene expression PCs before training GVF, which is in part known to remove trans effects, such as driver mutations in oncogenes, that cause transcriptome-wide changes in gene expression [2]. We motivate this regression in more detail in response to the reviewer’s penultimate comment. To evaluate GVF’s performance depending on known driver mutation status, we will add a performance breakdown by PAM50 subtype to the appendix, as subtypes closely correspond to HER2, ER, and PR mutation status.
>
> [1] J. Kvamme, M. B. Badsha, E. A. Martin, J. Wu, X. Wang, and A. Q. Fu, “Types of cis- and trans-gene regulation of expression quantitative trait loci across human tissues,” Sept. 13, 2023, bioRxiv. doi: 10.1101/2022.01.24.477617.
>
> [2] T. O. Nielsen et al., “A Comparison of PAM50 Intrinsic Subtyping with Immunohistochemistry and Clinical Prognostic Factors in Tamoxifen-Treated Estrogen Receptor–Positive Breast Cancer,” Clin Cancer Res, vol. 16, no. 21, pp. 5222–5232, Oct. 2010, doi: 10.1158/1078-0432.CCR-10-1282.
>
> **The authors mention that one of the major benefits of their method over others is its generalizability to new genes, such as fusion products. How can this be validated in TCGA data?**
>
> We thank the reviewer for identifying this opportunity. Most fusion products are expressed at levels more similar to the 5’ partner, so we will first quantify how well these correlate in this breast cancer cohort [1]. If so, we will consider samples that have fusion products and use GVF to predict expression by using mutations within 16 Mbp of the 5’ partner’s transcription start site and quantify how well this correlates with fusion expression levels across the cohort. Especially for inter-chromosomal fusions, this will not provide GVF an accurate input for the mutations downstream of the 3’ partner. For an initial experiment this should still provide a reasonable lower bound for how well GVF may perform in this setting. We will also add a brief discussion for how this may be improved: the main issue is that the downstream mutations will be missing in the initial approach. This can be corrected by, for example, querying mutations 8 Mbp downstream of the 3’ partner’s transcription termination site and concatenating these with the upstream mutations, and then making the downstream mutations’ positions relative to the 3’ partner’s TSS using the length of the fusion product.
>
> [1] Q. Gao et al., “Driver Fusions and Their Implications in the Development and Treatment of Human Cancers,” Cell Reports, vol. 23, no. 1, pp. 227-238.e3, Apr. 2018, doi: 10.1016/j.celrep.2018.03.050.
>
> **If Borzoi cannot be fine-tuned due to resource constraints, it should not be included. A Pearson correlation of 0.0043, which is most likely not statistically significant, is not very informative.**
>
> We agree that Borzoi’s performance is practically insignificant. We will clarify that the purpose of including Borzoi is to: (1) show that GVF’s performance is not solely from Borzoi gene promoter embeddings, since these are used as an input and (2) demonstrate that, as expected from results in non-cancer, Borzoi without fine-tuning does not generalize to breast tumor genomes.
>
> **A correlation number by itself, especially when it is only 0.22, is not very meaningful. A couple of examples of how GVF can find hallmark genes with significant predicted and observed differential expression will be helpful.**
>
> We appreciate the reviewer’s suggestion to better contextualize the Pearson correlation metric. We will clarify that a Pearson’s ρ of 0.21 is high relative to comparable baselines. While not a perfect comparison, models using germline variants to predict gene expression in normal blood report Pearson correlations of 0.29-0.37 in seen genes [1,2].
>
> (continued)

---

> ### Author Response · Authors · 2025-11-26
> **Reply to review (4/5)**
>
> To evaluate via differential expression, we would need to construct a mutation-derived phenotype to group samples. However, this is non-trivial because, across all 16 Mbp regions centered on genes, there is not a single gene where the fraction of samples with no mutations is between 1% and 99%. Across all genes, the maximum number of samples without any mutations in a 16 Mbp query region is just four, making it very difficult to do any analysis based on grouping samples by presence/absence of mutations. We would likely need to functionally annotate the non-coding somatic mutations around hallmark genes using an independent tool and then come up with a method to aggregate those scores for splitting patients into two or more groups. The conclusions we could draw from this analysis would strongly depend on the quality of the independent tool so we would have to assess this carefully.
>
> An alternative we can explore that still helps contextualize the model across hallmark genes is to measure how well hallmark breast cancer genes are predicted and whether there is an enrichment for hallmark genes in the distribution of Pearson correlations, e.g., via gene set enrichment analysis.
>
> [1] E. R. Gamazon et al., “A gene-based association method for mapping traits using reference transcriptome data,” Nat Genet, vol. 47, no. 9, pp. 1091–1098, Sept. 2015, doi: 10.1038/ng.3367.
>
> [2] R. Rastogi, A. J. Reddy, R. Chung, and N. M. Ioannidis, “Fine-tuning sequence-to-expression models on personal genome and transcriptome data,” Sept. 25, 2024, bioRxiv. doi: 10.1101/2024.09.23.614632.
>
> **The authors predict “in silico purified” cancer expression (InstaPrism) and also regress out the top 10 PCs before z-scoring; both steps can remove real signal or introduce target noise/bias. There are genuine differences between subtypes, which seems to be wiped out (Figs. 4A and B).**
>
> We agree with the reviewer that both *in silico* purification and regressing out the top 10 PCs can potentially remove some genuine biological variation from the data. The purity of solid tumor biopsies, or the percent of cells that are cancerous, can vary dramatically across a cohort. In this breast cohort, it ranged from 15% to ~100%. This variation isn’t necessarily due to contamination, but rather capturing surrounding normal tissue and tumors that are distinct clusters of cancer cells interspersed with normal cells. Many deconvolution methods like InstaPrism exist, with perhaps Newman et al. 2015 as the seminal paper with over 11k citations [1]. Over the years, deconvolution, or *in silico* purification, methods have led to a better understanding of the tumor microenvironment. The goal of our study was to model how somatic mutations affect somatic gene expression. While studying how somatic mutations affect the gene expression of neighboring cells in the tumor microenvironment is a very interesting question, it was out of scope for this study.
>
> Regarding regressing or conditioning on the top expression PCs, this is best practice for both somatic and germline eQTL studies, as well as germline sequence-to-expression models [2,3,4,5,6]. This removes systematic biases in gene expression caused by batch effects in sample preparation and sequencing [7]. These batch effects are so impactful that the field has converged on using this approach to remove them, despite its imprecision. Following best practices, we chose 10 expression PCs by inspection of the PC scree plot, which will be added to the appendix.
>
> [1] A. M. Newman et al., “Robust enumeration of cell subsets from tissue expression profiles,” Nat Methods, vol. 12, no. 5, pp. 453–457, May 2015, doi: 10.1038/nmeth.3337.
>
> [2] W. Zhang et al., “A global transcriptional network connecting noncoding mutations to changes in tumor gene expression,” Nat Genet, vol. 50, no. 4, pp. 613–620, Apr. 2018, doi: 10.1038/s41588-018-0091-2.
>
> [3] S. Drusinsky, S. Whalen, and K. S. Pollard, “Deep-learning prediction of gene expression from personal genomes,” July 27, 2024, bioRxiv. doi: 10.1101/2024.07.27.605449.
>
> [4] R. Rastogi, A. J. Reddy, R. Chung, and N. M. Ioannidis, “Fine-tuning sequence-to-expression models on personal genome and transcriptome data,” Sept. 25, 2024, bioRxiv. doi: 10.1101/2024.09.23.614632.
>
> [5] A. E. Spiro et al., “A scalable approach to investigating sequence-to-function predictions from personal genomes,” July 29, 2025, bioRxiv. doi: 10.1101/2025.02.21.639494.
>
> [6] E. Vaz, L. Wang, J. Galvin, R. Keener, and A. Battle, “Fine-tuning sequence to function deep learning models on large-scale proteomic data improves the accuracy of variant effect prediction,” Sept. 27, 2025, bioRxiv. doi: 10.1101/2025.09.26.678908.
>
> [7] H. J. Zhou, L. Li, Y. Li, W. Li, and J. J. Li, “PCA outperforms popular hidden variable inference methods for molecular QTL mapping,” Genome Biol, vol. 23, no. 1, p. 210, Oct. 2022, doi: 10.1186/s13059-022-02761-4.
>
> (continued)

---

> > ### Author Response · Authors · 2025-11-26
> > **Reply to review (5/5)**
> >
> > **The survival analysis results use linear probes and are strongest only in luminal A (where are Luminal B and Her2?); robustness across subtypes and cohorts remains untested. I would also suggest reporting the Cox HRs.**
> >
> > We thank the reviewer for the opportunity to clarify this. The third experiment is designed to evaluate whether the model has learned meaningful biology regarding the effect of non-coding mutations on gene expression. We fit a comprehensive sweep of Cox proportional hazard models using all features listed in Table 1 across all subtypes as well as the full cohort. We will include a comprehensive table in the appendix. We observed that standard deviations of the concordance indices were quite high, sometimes overlapping with 0.5, random performance. When all mutation-based models were within 1 standard deviation of random, we omitted that task from the table. This high variance in performance was likely driven by sample size limitations. We unfortunately cannot report interpretable hazard ratios since we used PCA for dimensionality reduction prior to fitting CoxPH models. However, we will add statistical tests between models, replace standard deviations with 95% CIs, and rephrase our claims to state GVF achieved competitive performance vs. advantaged baselines with trending improvements over mutation-only baselines.

---

### Official Review · Reviewer_1KZA · 2025-10-28

**Soundness:** 3
**Presentation:** 4
**Contribution:** 3
**Rating:** 6
**Confidence:** 4

**Summary:**

This paper introduces GenVarFormer (GVF), a novel transformer-based architecture designed to predict gene expression from sparse, long-range somatic mutations in cancer. The key innovation is a model that processes only the mutated sites and their local DNA context within a very large window (up to 16 Mbp), omitting the vast intermediate sequence to achieve computational efficiency. The authors evaluate GVF on 864 breast cancer samples from TCGA, demonstrating a strong improvement in gene expression prediction over standard models. Furthermore, they claim that patient embeddings derived from GVF are more informative for predicting patient survival in the Luminal A breast cancer subtype than the ground-truth gene expression data the model was trained on.

**Strengths:**

- The core idea of modeling sparse, long-range genomic interactions by omitting the intermediate sequence is highly original and significant.

- The work demonstrates high technical quality. The authors implemented several non-trivial engineering solutions (nested tensors for efficiency, a custom RoPE kernel for arbitrarily positioned tokens, and a bin-packing sampler) that were critical for making their approach practical and efficient.

- The paper is written with outstanding clarity. The motivation, methods, and results are presented logically and are easy to follow. The authors successfully bridge the gap between a complex biological problem and a sophisticated machine learning solution.

- The authors include a strong set of baselines, including standard bioinformatics methods (lasso on hotspots), state-of-the-art sequence models (Borzoi), and simple but effective heuristics (mean expression per subtype). The ablation on window size and the inclusion of an untrained model for the clinical tasks strengthen the paper's conclusions.

**Weaknesses:**

- The study relies on a single dataset of 864 samples. While TCGA is a high-quality resource, this sample size is small for deep learning and limits the statistical power of the findings, especially for the survival analysis. The lack of validation on an external cohort makes it difficult to assess the true generalizability of the model and its findings.

- The claim that GVF embeddings are superior to gene expression for survival prediction is a key result, but it should be tempered. This finding only holds for one of two tested cancer subtypes (Luminal A) and is not statistically robust, as indicated by the large and overlapping confidence intervals. This high variance is likely a direct consequence of the small dataset.

- While the 26-fold improvement over the lasso baseline is striking, this baseline's performance is near zero. A more sober and arguably fairer comparison is to the "mean subtype" baseline, where GVF still shows a strong ~3-fold improvement. Highlighting the 26-fold figure in the abstract feels like an overstatement of the practical performance gain.

**Questions:**

- The reliance on a single cohort is the primary limitation of this work. Can the authors comment on the availability of other public datasets with paired whole-genome and RNA sequencing data that could be used for external validation?

- Given the large and overlapping confidence intervals for the survival prediction results in Luminal A, have the authors performed a formal statistical test to assess if this difference is significant? Could the authors tone down the claims in the abstract and conclusion to better reflect this uncertainty?

- The "Advantaged Features" (e.g., gene expression, copy number) provide a strong performance ceiling for the clinical prediction tasks. Have the authors considered a model that combines their mutation-derived GVF embeddings with these features? While the goal of the paper is to demonstrate the power of mutation data alone, showing that GVF provides complementary information could further strengthen the paper's contribution.

---

> ### Author Response · Authors · 2025-11-26
> **Reply to review**
>
> We are grateful for the reviewer’s suggestions to strengthen this work and more effectively communicate the conclusions. We have responded to each weakness/question below.
>
> **The study relies on a single dataset of 864 samples. While TCGA is a high-quality resource, this sample size is small for deep learning and limits the statistical power of the findings, especially for the survival analysis. The lack of validation on an external cohort makes it difficult to assess the true generalizability of the model and its findings. … Can the authors comment on the availability of other public datasets with paired whole-genome and RNA sequencing data that could be used for external validation?**
>
> We thank the reviewer for this suggestion. (Reviewer 2 and 4 had very similar comments so we have written a unified response.) We will add a discussion on the need for more public, independent datasets with paired WGS and RNA-seq from breast cancer. We will also explore adding an evaluation on data from Hartwig Medical Foundation, the most comparable independent dataset to our knowledge. However, it would not be unsurprising if performance dropped, since Hartwig consists of almost exclusively metastatic tumors, whereas TCGA (the training data) is almost exclusively primary tumors. Generalization across domains is out of scope for this work.
>
> **The claim that GVF embeddings are superior to gene expression for survival prediction is a key result, but it should be tempered. This finding only holds for one of two tested cancer subtypes (Luminal A) and is not statistically robust, as indicated by the large and overlapping confidence intervals. This high variance is likely a direct consequence of the small dataset. … Given the large and overlapping confidence intervals for the survival prediction results in Luminal A, have the authors performed a formal statistical test to assess if this difference is significant? Could the authors tone down the claims in the abstract and conclusion to better reflect this uncertainty?**
>
> The reviewers point is well taken and we will temper the claims in the abstract and conclusion to reflect the sample size limitations. We will test for significance by bootstrapping the results from experiment 3 and discuss how sample size limitations are likely driving the large and often overlapping standard deviations (replaced with 95% CIs after bootstrapping) across all models. We will temper claims to highlight competitive performance vs. advantaged baselines, and trending improvements over mutation-only baselines.
>
> **While the 26-fold improvement over the lasso baseline is striking, this baseline's performance is near zero. A more sober and arguably fairer comparison is to the "mean subtype" baseline, where GVF still shows a strong ~3-fold improvement. Highlighting the 26-fold figure in the abstract feels like an overstatement of the practical performance gain.**
>
> We appreciate this recommendation to highlight a more sober comparison and will include the 3-fold improvement in the abstract. The purpose of measuring Pearson correlation across unseen samples and genes is to quantify how informative a model is for identifying how mutations affect gene expression, and the mean of each subtype has no value for this since it doesn’t use mutations. We will clarify that the mean subtype baseline was included to verify that GVF made non-trivial predictions.
>
> **The "Advantaged Features" (e.g., gene expression, copy number) provide a strong performance ceiling for the clinical prediction tasks. Have the authors considered a model that combines their mutation-derived GVF embeddings with these features? While the goal of the paper is to demonstrate the power of mutation data alone, showing that GVF provides complementary information could further strengthen the paper's contribution.**
>
> We agree with the reviewer’s suggestion. While there is some evidence that GVF embeddings are complementary to the advantaged features based on differences in performance, ensembling them with gene expression and copy number would more clearly test whether the embeddings provide orthogonal information. We will concatenate the features together and add the ensemble result to the manuscript.

---

### Official Review · Reviewer_dm4o · 2025-10-30

**Soundness:** 2
**Presentation:** 2
**Contribution:** 2
**Rating:** 2
**Confidence:** 3

**Summary:**

GenVarFormer is a dl model that utilizes somatic mutation information surrounding a gene to predict the expression of interest. The model is a transformer with its input integrate various sources of information: mutations, flanking DNA sequences and gene embeddings. Authors train the model on 864 samples with paired WGS data (used to derive mutations) and RNA-seq (labels). They performed two type of experiments. Experiment 1 evaluates the regression performance of GVF versus Lasso hotspot models, Borzoi and a mean expression of subtypes. Experiment 2 use the model's embedding to cluster the samples into cell types.

**Strengths:**

The model implementation seems reasonable, which has adopted recent technologies such as Gradient Aligned Regression as loss and several technique to generate embeddings of mutations and variant allele fractions.

**Weaknesses:**

The experiments are relatively weak. The aim is to predict the gene expression using somatic mutations. The prediction of cell type mean is already 0.075, while GVF is 0.219, which is very low for Pearson correlation. Also, authors have regresses out the 10 principle components out in the training. This means that the model is actually trying to learn the residuals, while the most interesting variations may lie in the 10 regressed out PCs. I did not quite get the logic of experiment 2, the method should provide added value on top of cancer subtype classification. If one can easily infer the cancer subtypes from gene expression, what is the benefit of using a transformer to predict the same thing using the mutation information, which are much harder to derive and prone to errors.

I think the authors might consider
1. include two or three independent datasets as test data
2. try to include classic regression models such as simple linear regression model (y ~ cancer subtype + patient_id + \epsilon) and Gaussian process regression
3. not regressing out the first 10 PCs in experiment 1
4. use effect size or explained variance type of metric instead of Pearson correlation in experiment 1
5. show the biological relevance of the model prediction, e.g. ranking the input mutations and the top mutations are known driver mutations. the cancer subtype prediction is an easy task given mutations or gene expressions

**Questions:**

1. what is the effect size of residuals that GVF is trying to learn? I feel there might be not too much variance left after regressing out the first 10 PCs.
2. Can the Flanking DNA provide some information of driver mutations?

---

> ### Author Response · Authors · 2025-11-26
> **Reply to review (1/2)**
>
> We thank the reviewer for their helpful questions regarding contextualizing the performance metrics, our preprocessing methodology, and the purpose of the experiments. We have responded to each weakness/question below.
>
> **The prediction of cell type mean is already 0.075, while GVF is 0.219, which is very low for Pearson correlation.**
>
> We will clarify that the mean gene expression of each subtype provides no value for understanding how mutations affect gene expression, and is merely shown to verify that GVF made non-trivial predictions. Performance of the comparable baseline models demonstrate that predicting gene expression is a difficult task and a Pearson of 0.219 is substantial. To our knowledge, performance of models predicting gene expression in unseen genes and exclusively from mutations has not been reported before. Models using germline variants to predict expression in normal blood report Pearson correlations of 0.28-0.37 in seen genes [1,2].
>
> [1] E. R. Gamazon et al., “A gene-based association method for mapping traits using reference transcriptome data,” Nat Genet, vol. 47, no. 9, pp. 1091–1098, Sept. 2015, doi: 10.1038/ng.3367.
>
> [2] R. Rastogi, A. J. Reddy, R. Chung, and N. M. Ioannidis, “Fine-tuning sequence-to-expression models on personal genome and transcriptome data,” Sept. 25, 2024, bioRxiv. doi: 10.1101/2024.09.23.614632.
>
> **Also, authors have regresses out the 10 principle components out in the training. This means that the model is actually trying to learn the residuals, while the most interesting variations may lie in the 10 regressed out PCs. … (3) Consider not regressing out the first 10 PCs in experiment 1. … (6) What is the effect size of residuals that GVF is trying to learn? I feel there might be not too much variance left after regressing out the first 10 PCs.**
>
> Regressing the top expression principal components (PC) is best practice for both somatic and germline eQTL studies, as well as germline sequence-to-expression models. This removes systematic biases in gene expression caused by batch effects in sample preparation and sequencing [1]. We chose to use the top 10 expression PCs by visually identifying the elbow of the PC scree plot, which we will add to the appendix.
>
> [1] H. J. Zhou, L. Li, Y. Li, W. Li, and J. J. Li, “PCA outperforms popular hidden variable inference methods for molecular QTL mapping,” Genome Biol, vol. 23, no. 1, p. 210, Oct. 2022, doi: 10.1186/s13059-022-02761-4.
>
> **I did not quite get the logic of experiment 2, the method should provide added value on top of cancer subtype classification. If one can easily infer the cancer subtypes from gene expression, what is the benefit of using a transformer to predict the same thing using the mutation information, which are much harder to derive and prone to errors.**
>
> We will clarify that experiment 2 is a case study to evaluate whether the structure of a GVF encoder is effective for encoding mutations, rather than to show GVF can classify subtypes. Prior work has found that breast cancer subtypes display distinct, genome-wide mutational profiles and are driven by different driver mutations [1,2,3]. Thus, we should expect an effective representation of tumor genomes to roughly cluster by subtype. We use an untrained GVF encoder to evaluate this because it corresponds to dimensionality reduction via random matrix projection. We will better emphasize this so it is clear that experiment 2 assesses the value of just the mutation encoder without the downstream transformer. A final practical benefit is that hotspot algorithms have many parameters to tune, whereas an untrained GVF encoder only requires choosing a dimensionality.
>
> [1] G. Perry et al., “Divergence of mutational signatures in association with breast cancer subtype,” Molecular Carcinogenesis, vol. 61, no. 11, pp. 1056–1070, 2022, doi: 10.1002/mc.23461.
>
> [2] P. Polak et al., “A mutational signature reveals alterations underlying deficient homologous recombination repair in breast cancer,” Nat Genet, vol. 49, no. 10, pp. 1476–1486, Oct. 2017, doi: 10.1038/ng.3934.
>
> [3] W. A. M. E. Schrijver et al., “Mutation Profiling of Key Cancer Genes in Primary Breast Cancers and Their Distant Metastases,” Cancer Res, vol. 78, no. 12, pp. 3112–3121, June 2018, doi: 10.1158/0008-5472.CAN-17-2310.
>
> **I think the authors might consider**
>
> **1. include two or three independent datasets as test data**
>
> We will add a discussion on the need for more public, independent datasets with paired WGS and RNA-seq from breast cancer. We will also explore adding an evaluation on Hartwig, the most comparable independent dataset to our knowledge. However, it would not be unsurprising if performance dropped, since Hartwig consists of almost exclusively metastatic tumors, whereas TCGA (the training data) is almost exclusively primary tumors. Generalization across domains is out of scope for this work.

---

> ### Author Response · Authors · 2025-11-26
> **Reply to review (2/2)**
>
> **2. try to include classic regression models such as simple linear regression model (y ~ cancer subtype + patient_id + \epsilon) and Gaussian process regression**
>
> We will clarify that linear models are included in the paper. A lasso model is used as a baseline for gene expression prediction (experiment 1) and all models in experiment 3 are either linear Cox proportional hazards models, or regularized logistic regression models. The Gaussian process regression suggestion is well taken, and we will explore fitting these for experiment 3 in case they lead to any different conclusions.
>
> **3. consider not regressing out the first 10 PCs in experiment 1**
>
> See above, where we responded to the corresponding comment/weakness.
>
> **4. use effect size or explained variance type of metric instead of Pearson correlation in experiment 1**
>
> We will add $R^2$ metrics for all models to the appendix. We agree that it is helpful to have another metric presented and expect the results to be consistent with the current conclusions about GVF’s quality of representations for how mutations affect gene expression.
>
> **5. show the biological relevance of the model prediction, e.g. ranking the input mutations and the top mutations are known driver mutations. the cancer subtype prediction is an easy task given mutations or gene expressions**
>
> We will clarify that experiment 2 and 3 test aspects of biological relevance, as biologically uninformative embeddings should have little concordance with survival, tumor subtype, or stage. We agree that cancer subtype is generally an easy task, especially when gene expression or gene copy number are available. However, the benchmark demonstrates that discerning subtype from non-coding mutations alone is more challenging. We show that GVF embeddings are the most predictive mutation-based representation for these phenotypes, and in some cases are comparable to non-mutation molecular signals such as gene expression and copy number.
>
> **Questions**
>
> **6. what is the effect size of residuals that GVF is trying to learn? I feel there might be not too much variance left after regressing out the first 10 PCs.**
>
> See above, where we responded to the corresponding comment/weakness.
>
> **7. Can the Flanking DNA provide some information of driver mutations?**
>
> We will add clarification motivating the inclusion of each mutation’s flanking DNA. Flanking DNA is highly informative for characterizing driver mutations, as the functional impact of a variant is highly dependent on its local context. Specifically, flanking DNA (1) informs the regulatory syntax (e.g., transcription factor binding motifs, typically ≤30 bp [1]) that a mutation may disrupt or create, (2) captures mutational signatures (e.g., APOBEC or UV damage), which are defined by trinucleotide contexts [2], and (3) can reflect mutational biases within coding regions [3,4,5]. Without this surrounding sequence information, functional drivers and background passengers are likely indistinguishable, and we would expect predictive performance would degrade significantly without it.
>
> [1] J. Vierstra et al., “Global reference mapping of human transcription factor footprints,” Nature, vol. 583, no. 7818, pp. 729–736, July 2020, doi: 10.1038/s41586-020-2528-x.
>
> [2] L. B. Alexandrov et al., “The repertoire of mutational signatures in human cancer,” Nature, vol. 578, no. 7793, pp. 94–101, Feb. 2020, doi: 10.1038/s41586-020-1943-3.
>
> [3] H. Benisty, M. Weber, X. Hernandez-Alias, M. H. Schaefer, and L. Serrano, “Mutation bias within oncogene families is related to proliferation-specific codon usage,” Proceedings of the National Academy of Sciences, vol. 117, no. 48, pp. 30848–30856, Dec. 2020, doi: 10.1073/pnas.2016119117.
>
> [4] L. Zapata, O. Pich, L. Serrano, F. A. Kondrashov, S. Ossowski, and M. H. Schaefer, “Negative selection in tumor genome evolution acts on essential cellular functions and the immunopeptidome,” Genome Biol, vol. 19, no. 1, p. 67, May 2018, doi: 10.1186/s13059-018-1434-0.
>
> [5] D. J. Hsu et al., “Arginine limitation drives a directed codon-dependent DNA sequence evolution response in colorectal cancer cells,” Science Advances, vol. 9, no. 1, p. eade9120, Jan. 2023, doi: 10.1126/sciadv.ade9120.

---

### Official Review · Reviewer_e2vU · 2025-11-02

**Soundness:** 2
**Presentation:** 1
**Contribution:** 1
**Rating:** 2
**Confidence:** 4

**Summary:**

The paper proposes a CNN-and-Transformer-based model for predicting gene expression from DNA, with the goal of untangling cancer-related alternations in expression stemming from DNA mutations.

**Strengths:**

The proposed tool is aimed at a difficult and important problem of predicting the effect of cancer driver mutations on gene expression and, downstream, on clinical variables such as survival, subtype, and stage. The paper goes beyond standard DNA-based Transformer but proposing a two-step pipeline involving mutations and gene sequence embedding, followed by a Transformer operating on the embeddings.

**Weaknesses:**

From a machine learning perspective alone, the proposed solution, involving embeddings and a Transformer, lacks substantial novelty. Unfortunately, the lack of details and broader context in describing the application (cancer mutations and their effect), as well as in placing the results in context of other bioinformatics approaches, do not show clear evidence that the proposed method substantially advances machine learning-based cancer bioinformatics.

The manuscript is missing some crucial information that would put the work in context, especially for a broader machine learning audience. For example, while the introduction highlights the sparsity of mutations along the DNA, it does not provide information about the distribution of distances of regulatory sites that may be affected by these mutations from the TSS of genes they regulate. Without this information, it is unclear whether the selected window size of 16M is sufficient, or necessary.

Further, within the 16M window, the method operates by focusing on smaller window around mutation sites. The experimental section is vague in terms of how are mutations selected – are all mutations present in TCGA within the window used, or some approach is used to select driver mutations?

It is also unclear how much DNA (how many bps) the method actually utilizes in making the prediction. The sentence “GVF uses a DNA context window of 16Mbp” should be rephrased: for a general machine learning audience, this is understood as indicating that the Transformer model in GVF takes DNA sequences that are 16Mbp long on input (the phrase “context window” has a precise technical meaning in LLM-related domain), while in fact it seems to take embeddings of 32bp sequences from each of the n mutations present within 16Mbp (in that sense, it is more similar to a RAG system, using LLM terms). It is not clear from the experimental section what are the typical number of mutations (i.e., how many 32bp windows are being embedded and presented to the Transformer model). For comparison, Borzoi model takes on input a full 0.5Mbp sequence of DNA.

The experimental results from evaluating the model are weak, and presented without much discussion. Predictive accuracy, as measured by Person’s rho, is very low (average of .21), but the utility of having a predictive model of expression changes with this level of accuracy is not discussed. In predicting expression changes from mutations, in a somewhat different experimental setup, Enformer and Borzoi achieve values mostly in the 0.4-0.6 range (see Liner et al, 2023, supplemental data/figures). Further, Fig 3B indicates that accuracy on “unseen genes and samples” is somewhat higher than for “unseen genes”, yet this surprising result is not discussed.

Results on predicting downstream clinical variables are also low (e.g. AUROC of 0.56 for early/late stage prediction), yet again are presented without discussion or context (e.g. what are the reference values for predicting overall/progression-free survival for these cancer types using existing established approaches).

**Questions:**

See weaknesses above.

---

> ### Author Response · Authors · 2025-11-26
> **Reply to review (1/4)**
>
> We thank the reviewer for their constructive comments and for pointing out critical opportunities to clarify and contextualize the paper. We have responded to each weakness/question below.
>
> **From a machine learning perspective alone, the proposed solution, involving embeddings and a Transformer, lacks substantial novelty. Unfortunately, the lack of details and broader context in describing the application (cancer mutations and their effect), as well as in placing the results in context of other bioinformatics approaches, do not show clear evidence that the proposed method substantially advances machine learning-based cancer bioinformatics.**
>
> Thank you for emphasizing the importance of communicating the novelty and broader impact of our work. We agree that while transformer architectures are well-established in general machine learning, their specific application to somatic non-coding bioinformatics represents a substantial departure from existing work. More broadly, to our knowledge, no machine learning model has been trained to learn the effects of mutations on gene expression in cancer. This is non-trivial, and the novelty of our work is primarily in the application and the technical advances required to implement it, rather than the architecture.
>
> Related ML approaches for non-coding variants (e.g., Evo 2, Borzoi, CADD) rely exclusively on evolutionary conservation across species (in the form of reference genomes) or human germline constraints [1,2,3]. These models inherently prioritize negative selection: purifying selection against deleterious alleles that are catastrophic to embryonic development or the function of any number of human cell types. In contrast, cancer genomes are largely unconstrained by germline selection, and are instead shaped by positive selection (clonal expansion) and distinct somatic mutational processes (e.g., replication timing biases, APOBEC activity) that reference- and germline-trained models are oblivious to [4,5,6].
>
> We will revise the manuscript to explicitly frame our contribution as addressing this specific gap where reference- and germline-based ML have never before been trained to learn the effects of non-coding mutations on gene expression in cancer. We show that learning the functional effects of somatic evolution, which distinctively differ from germline evolution, requires a dedicated architecture trained on somatic WGS data, rather than transfer learning from reference genomes.
>
> [1] G. Brixi et al., “Genome modeling and design across all domains of life with Evo 2,” Feb. 21, 2025, bioRxiv. doi: 10.1101/2025.02.18.638918.
>
> [2] J. Linder, D. Srivastava, H. Yuan, V. Agarwal, and D. R. Kelley, “Predicting RNA-seq coverage from DNA sequence as a unifying model of gene regulation,” Nat Genet, vol. 57, no. 4, pp. 949–961, Apr. 2025, doi: 10.1038/s41588-024-02053-6.
>
> [3] M. Schubach, T. Maass, L. Nazaretyan, S. Röner, and M. Kircher, “CADD v1.7: using protein language models, regulatory CNNs and other nucleotide-level scores to improve genome-wide variant predictions,” Nucleic Acids Res, vol. 52, no. D1, pp. D1143–D1154, Jan. 2024, doi: 10.1093/nar/gkad989.
>
> [4] I. Martincorena et al., “Universal Patterns of Selection in Cancer and Somatic Tissues,” Cell, vol. 171, no. 5, pp. 1029-1041.e21, Nov. 2017, doi: 10.1016/j.cell.2017.09.042.
>
> [5] H. Hornshøj et al., “Pan-cancer screen for mutations in non-coding elements with conservation and cancer specificity reveals correlations with expression and survival,” npj Genomic Med, vol. 3, no. 1, p. 1, Jan. 2018, doi: 10.1038/s41525-017-0040-5.
>
> [6] H. Carter et al., “Cancer-Specific High-Throughput Annotation of Somatic Mutations: Computational Prediction of Driver Missense Mutations,” Cancer Res, vol. 69, no. 16, pp. 6660–6667, Aug. 2009, doi: 10.1158/0008-5472.CAN-09-1133.
>
> **The manuscript is missing some crucial information that would put the work in context, especially for a broader machine learning audience. For example, while the introduction highlights the sparsity of mutations along the DNA, it does not provide information about the distribution of distances of regulatory sites that may be affected by these mutations from the TSS of genes they regulate. Without this information, it is unclear whether the selected window size of 16M is sufficient, or necessary.**
>
> (continued)

---

> ### Author Response · Authors · 2025-11-26
> **Reply to review (2/4)**
>
> Thank you for highlighting the need to better motivate our input window size (16 Mbp) relative in the context of known biological phenomena. The broader goal our study contributes to is learning how somatic mutations affect gene expression, irrespective of genomic distance. However, for tractability, we focused on mutations surrounding each gene and found that predictive performance improved with the size of the query window up to the maximum we tested, 16 Mbp. Generally speaking, a genome-wide window would be ideal since it is well understood that somatic mutations can affect genes that are even on separate chromosomes (e.g., TP53 mutations causing broad dysregulation of genes controlling cell cycle and apoptosis [1]). Using a 16 Mbp window around genes will not capture these trans-effects across chromosomes, but may still capture many other gene regulatory phenomena affected by more proximal mutations. In a healthy germline context, cis-regulatory elements are typically located within 1 Mbp of the TSS. This encompasses both the typical size of evolutionarily conserved topologically associating domains (TAD) and cell-type specific contact domains, which limit what genomic elements may come into physical contact [2,3]. However, many somatic-specific processes motivate a broader receptive field.
>
> We will update the manuscript to explicitly justify a 16 Mbp window based on three somatic-specific phenomena that extend beyond standard germline cis-regulatory distances:
> Unlike germline evolution, which preserves TADs, somatic genome instability frequently rearranges chromatin. Structural variants (SVs) can relocate enhancers from tens of megabases away to activate oncogenes (e.g., GFI1B or MYC), a phenomenon known as "enhancer hijacking". Enhancer hijacking has also been specifically observed in the TCGA breast cancer cohort [4,5]. Novel cis-regulatory interactions across much larger distances than normal can also occur from single nucleotide variants (SNV) and indels that disrupt TAD boundaries (e.g., by disabling insulators) [6].
> Tumors often exhibit global epigenetic remodeling, such as broad genic repression domains, large organized chromatin K9-modifications and hypomethylated blocks, which allow activation of cis-regulatory elements that are normally epigenetically silenced [7,8].
> A larger window allows the model to implicitly capture interactions between neighboring genes (e.g., a mutation in a transcription factor located 5 Mbp away affecting the target gene), approaching an "omnigenic" view of regulation that narrow windows miss [9].
>
> While a whole-genome context would be theoretically ideal to capture inter-chromosomal events (like gene fusions), it remains computationally intractable. We posit that 16 Mbp represents an effective trade-off: it is sufficiently large to capture the extended range of somatic cis-regulatory disruptions and regional chromatin remodeling, while remaining computationally feasible.
>
> [1] K. Sasaki, S. Takahashi, K. Ouchi, Y. Otsuki, S. Wakayama, and C. Ishioka, “Different impacts of TP53 mutations on cell cycle-related gene expression among cancer types,” Sci Rep, vol. 13, no. 1, p. 4868, Mar. 2023, doi: 10.1038/s41598-023-32092-8.
>
> [2] J. R. Dixon et al., “Topological domains in mammalian genomes identified by analysis of chromatin interactions,” Nature, vol. 485, no. 7398, pp. 376–380, May 2012, doi: 10.1038/nature11082.
>
> [3] S. S. P. Rao et al., “A 3D Map of the Human Genome at Kilobase Resolution Reveals Principles of Chromatin Looping,” Cell, vol. 159, no. 7, pp. 1665–1680, Dec. 2014, doi: 10.1016/j.cell.2014.11.021.
>
> [4] K. L. Mortenson et al., “3D genomic analysis reveals novel enhancer-hijacking caused by complex structural alterations that drive oncogene overexpression,” Nat Commun, vol. 15, no. 1, p. 6130, July 2024, doi: 10.1038/s41467-024-50387-w.
>
> [5] Y. Xi et al., “Super-enhancer-hijacking RBBP7 potentiates metastasis and stemness of breast cancer via recruiting NuRD complex subunit LSD1,” J Transl Med, vol. 23, no. 1, p. 266, Mar. 2025, doi: 10.1186/s12967-025-06270-3.
>
> [6] C. Tubbs, M. L. Benton, E. McArthur, J. A. Capra, and D. M. Ruderfer, “Identifying deleterious noncoding variation through gain and loss of CTCF binding activity,” The American Journal of Human Genetics, vol. 112, no. 4, pp. 892–902, Apr. 2025, doi: 10.1016/j.ajhg.2025.02.009.
>
> [7] D. Zhao et al., “Broad genic repression domains signify enhanced silencing of oncogenes,” Nat Commun, vol. 11, no. 1, p. 5560, Nov. 2020, doi: 10.1038/s41467-020-18913-8.
>
> [8] N. Besselink et al., “The genome-wide mutational consequences of DNA hypomethylation,” Sci Rep, vol. 13, no. 1, p. 6874, Apr. 2023, doi: 10.1038/s41598-023-33932-3.
>
> [9] E. A. Boyle, Y. I. Li, and J. K. Pritchard, “An Expanded View of Complex Traits: From Polygenic to Omnigenic,” Cell, vol. 169, no. 7, pp. 1177–1186, June 2017, doi: 10.1016/j.cell.2017.05.038.
>
> (continued)

---

> ### Author Response · Authors · 2025-11-26
> **Reply to review (3/4)**
>
> **Further, within the 16M window, the method operates by focusing on smaller window around mutation sites. The experimental section is vague in terms of how are mutations selected – are all mutations present in TCGA within the window used, or some approach is used to select driver mutations?**
>
> Thank you for pointing out this ambiguity. We will clarify in the manuscript that all mutations in the window are used and update the model schematic (Fig. 1) to better illustrate this.
>
> **It is also unclear how much DNA (how many bps) the method actually utilizes in making the prediction. The sentence “GVF uses a DNA context window of 16Mbp” should be rephrased: for a general machine learning audience, this is understood as indicating that the Transformer model in GVF takes DNA sequences that are 16Mbp long on input (the phrase “context window” has a precise technical meaning in LLM-related domain), while in fact it seems to take embeddings of 32bp sequences from each of the n mutations present within 16Mbp (in that sense, it is more similar to a RAG system, using LLM terms). It is not clear from the experimental section what are the typical number of mutations (i.e., how many 32bp windows are being embedded and presented to the Transformer model). For comparison, Borzoi model takes on input a full 0.5Mbp sequence of DNA.**
>
> Thank you for raising this ambiguity to our attention. We will clarify this important information about the input data:
> - Figure 2A is the distribution of mutations per gene-sample pair (i.e., per instance) for the 524 kb dataset (matching Borzoi). We will add another plot showing the number of mutations per gene-sample pair in the 16 Mbp dataset to the appendix, adding a bar for the median number of mutations, 29, and include this value in the figure caption.
> - The flanking DNA is 32 bp on either side for a total of 64 bp per mutation. This was chosen to be large enough to encompass transcription factor binding motifs.
> - This corresponds to a median sequence length of 1,856 per input, ignoring the alternate allele.
> - GVF also uses the indel length and variant allele fraction of each mutation.
>
> We will also rephrase all mentions of “context window” to avoid confusion given its meaning in the LLM domain. We agree that this will greatly help readers understand our work by better considering audiences that are familiar with LLMs and biological sequence models like Borzoi.
>
> **The experimental results from evaluating the model are weak, and presented without much discussion. Predictive accuracy, as measured by Person’s rho, is very low (average of .21), but the utility of having a predictive model of expression changes with this level of accuracy is not discussed. In predicting expression changes from mutations, in a somewhat different experimental setup, Enformer and Borzoi achieve values mostly in the 0.4-0.6 range (see Liner et al, 2023, supplemental data/figures). Further, Fig 3B indicates that accuracy on “unseen genes and samples” is somewhat higher than for “unseen genes”, yet this surprising result is not discussed.**
>
> We will clarify that the purpose of measuring Pearson correlation across unseen samples and genes is to quantify how informative each model is for identifying how mutations affect gene expression. A Pearson’s ρ of 0.21 is high relative to comparable baselines. While not a perfect comparison, models using germline variants to predict gene expression in normal blood report Pearson correlations of 0.28-0.37 in seen genes; these models do not generalize to unseen genes [1,2]. We also included Borzoi (which arguably supersedes Enformer) as a baseline and showed it had zero predictive value in this experimental setup. Fine-tuning Borzoi is out of scope since it is computationally intractable with our resources; recent work fine-tuning Borozi on paired WGS and proteomic data from blood required far more resources than we have and still had to restrict the DNA context window from 524 kb to 49 kb [3].
>
> We did not reference the published Pearson metrics from Enformer or Borzoi in their respective papers because the experimental setups are biologically incomparable to our study. We will acknowledge this in our discussion, emphasizing that our study focuses on evaluating our models ability to learn biologically relevant differences across individuals. Evaluations of Enformer and Borzoi were across genes or cell types, or to concordance with fine-mapped eQTLs. The latter are the effects of single variants in isolation, taken out of the personalized genomic context that has been shown to confuse these models [4,5].
>
> (continued)

---

> > ### Author Response · Authors · 2025-11-26
> > **Reply to review (4/4)**
> >
> > Regarding the higher performance on unseen genes, we agree that this is surprising. We will clarify that this could be due to splitting our genes by chromosome, which was necessary to avoid data leakage since some genes have overlapping coordinates. There are known patterns of mutations across chromosomes in breast cancer [6]. This may result in systematic differences across chromosomes for the effects of mutations on gene expression. We will add an ablation study using different held-out chromosomes and the same hyperparameters to verify whether this is the case. If true, we should see a different ratio in performance between seen vs. unseen genes. If not, this will rule out that chromosome-specific effects are causing this observation.
> >
> > [1] E. R. Gamazon et al., “A gene-based association method for mapping traits using reference transcriptome data,” Nat Genet, vol. 47, no. 9, pp. 1091–1098, Sept. 2015, doi: 10.1038/ng.3367.
> >
> > [2] R. Rastogi, A. J. Reddy, R. Chung, and N. M. Ioannidis, “Fine-tuning sequence-to-expression models on personal genome and transcriptome data,” Sept. 25, 2024, bioRxiv. doi: 10.1101/2024.09.23.614632.
> >
> > [3] E. Vaz, L. Wang, J. Galvin, R. Keener, and A. Battle, “Fine-tuning sequence to function deep learning models on large-scale proteomic data improves the accuracy of variant effect prediction,” Sept. 27, 2025, bioRxiv. doi: 10.1101/2025.09.26.678908.
> >
> > [4] C. Huang et al., “Personal transcriptome variation is poorly explained by current genomic deep learning models,” Nat Genet, vol. 55, no. 12, pp. 2056–2059, Dec. 2023, doi: 10.1038/s41588-023-01574-w.
> >
> > [5] A. Sasse et al., “Benchmarking of deep neural networks for predicting personal gene expression from DNA sequence highlights shortcomings,” Nat Genet, vol. 55, no. 12, pp. 2060–2064, Dec. 2023, doi: 10.1038/s41588-023-01524-6.
> >
> > [6] D. C. Koboldt et al., “Comprehensive molecular portraits of human breast tumours,” Nature, vol. 490, no. 7418, pp. 61–70, Oct. 2012, doi: 10.1038/nature11412.
> >
> > **Results on predicting downstream clinical variables are also low (e.g. AUROC of 0.56 for early/late stage prediction), yet again are presented without discussion or context (e.g. what are the reference values for predicting overall/progression-free survival for these cancer types using existing established approaches).**
> >
> > The reviewer’s point is well taken. We will clarify that the goal of this experiment is not to outperform existing best methods for clinical tasks, but to evaluate whether the model has learned meaningful biology regarding the effect of non-coding mutations on gene expression. To our knowledge, while coding mutations are relatively well-studied, there are no published methods for these tasks that quantify the predictive value of non-coding mutations. In addition, most prior work has not evaluated performance in each subtype as we show here. Breast cancer survival is often stratified by subtype, but there is a need for biomarkers that stratify survival within each subtype [1]. Classifying early vs. late stage tumors in this cohort was a challenging task that all models performed relatively poorly for, with the best AUROC of 0.63 for purified expression, though the small standard deviations permitted comparing the predictive value of different models.
> >
> > [1] S. Yadav, S. Zhou, B. He, Y. Du, and L. X. Garmire, “Deep learning and transfer learning identify breast cancer survival subtypes from single-cell imaging data,” Commun Med, vol. 3, no. 1, p. 187, Dec. 2023, doi: 10.1038/s43856-023-00414-6.

---

### Author Response · Authors · 2025-12-04
**Comment to Area Chair**

As the discussion period comes to a close, we would like to provide a final update on the status of our paper and the points raised during the review process.

In our initial response, we outlined a plan to address the reviewers' comments by re-training with different splits, adding Gaussian Process models for downstream tasks, and including an evaluation on another independent dataset we identified of substantial size. While we have initiated these experiments, the computational time required has exceeded the rebuttal window, and we were unable to include the final results before the deadline. We are committed to completing this work and incorporating the reviewers suggestions into the paper.

We also wanted to note that due to the broader disruptions regarding the anonymity leak during this cycle, we unfortunately did not have the opportunity to engage in a dialogue with the reviewers regarding our proposed plan. We understand this was a difficult cycle for everyone involved.

Despite the mixed scores, we believe the core contributions highlighted by Reviewer 1KZA and Fr92 warrant consideration. This work demonstrated substantial novelty by modeling sparse, long-range genomic interactions by omitting the intermediate sequence and required multiple non-trivial technical advances. The experimental results are strong in an important, challenging domain (understanding how non-coding somatic mutations affect gene expression) that is in need of improved modeling strategies. We hope the final decision can focus on the potential of these core ideas, noting that the empirical validation is currently being expanded as requested.

Thank you for your time and for overseeing the review process.

---

### Meta-Review · Area_Chair_2vR5 · 2026-01-07

**Summary:**

Reviewers broadly found the idea interesting, but raised major concerns that the current empirical evidence is not yet strong enough for ICLR, including modest predictive accuracy (average correlation around 0.21) and limited context/interpretation of what this accuracy enables.

A central theme across reviews is that key validations are missing: external cohort testing, stronger / fairer baselines and metrics, and clearer biological interpretability (e.g., concrete examples showing meaningful mutation-to-expression insights). One reviewer is supportive but asks to temper headline claims, emphasize fairer comparisons, and test complementarity with advantaged clinical features.

**Reviewer Concerns:**

### Addressed

- Overstatement / framing: Authors agree to tone down survival and “26x” claims, add bootstrap-based uncertainty (95% CIs), and highlight the more sober comparison vs the “mean subtype” baseline (about 3-fold).

- Complementarity with advantaged features: Authors agree to add an explicit combined-features result (concatenating GVF with gene expression / copy number variation).

- Baseline clarification: Authors note linear baselines already exist in the paper and plan to explore Gaussian Process regression for downstream tasks.


### Still outstanding (quite critical)

- No new results for the key asks: retraining with alternative splits, adding GP baselines, and evaluating an independent dataset were initiated but not completed within rebuttal due to compute/time.

- Predictive performance remains a sticking point: reviewers highlight low average correlation and ask for clearer utility/context and comparison to sequence models in related setups; downstream clinical prediction is also viewed as weak without proper context.

- Preprocessing and target definition concerns: regressing out top PCs and predicting in-silico purified expression may remove real biological signal or introduce bias; this needs deeper justification, clarifications and ablations.

- Scope of biology/features: missing epigenomic regulation (e.g., methylation), copy-number-driven effects, and other confounders; also requests for concrete case studies / hallmark examples and reporting additional survival details.

**Reviewer Scores:**

Reviewer e2vU: Likely unchanged at 2. Core concerns are about low accuracy and lack of contextualization; no new experimental results were added during rebuttal.

Reviewer dm4o: Likely unchanged at 2. Requested additional datasets, alternative baselines/metrics, and ablations (e.g., PC regression); these remain largely pending.

Reviewer Fr92: Likely unchanged at 4. Concerns span missing biology/features, questionable baselines, and limited robustness across subtypes/cohorts.

Reviewer 1KZA: Likely 5–6. The reviewer is already “marginally above threshold” but explicitly prioritizes external validation, toned-down claims, and showing complementarity with advantaged features (promised, but not yet demonstrated in results).

---

### Decision · Program_Chairs · 2026-01-26

Reject